# Quantitative Succinyl-Proteome Profiling of Turnip (*Brassica rapa* var. *rapa*) in Response to Cadmium Stress

**DOI:** 10.3390/cells11121947

**Published:** 2022-06-17

**Authors:** Xiong Li, Danni Yang, Yunqiang Yang, Guihua Jin, Xin Yin, Yan Zheng, Jianchu Xu, Yongping Yang

**Affiliations:** 1Germplasm Bank of Wild Species, Kunming Institute of Botany, Chinese Academy of Sciences, Kunming 650201, China; lixiong@mail.kib.ac.cn (X.L.); yangdanni@mail.kib.ac.cn (D.Y.); yangyunqiang@mail.kib.ac.cn (Y.Y.); jinguihua@mail.kib.ac.cn (G.J.); yinxin@mail.kib.ac.cn (X.Y.); zhengyan@mail.kib.ac.cn (Y.Z.); 2Yunnan Key Laboratory for Wild Plant Resources, Department of Economic Plants and Biotechnology, Kunming Institute of Botany, Chinese Academy of Sciences, Kunming 650201, China; 3Center for Mountain Futures, Kunming Institute of Botany, Chinese Academy of Sciences, Kunming 650201, China; 4Xishuangbanna Tropical Botanical Garden, Chinese Academy of Sciences, Xishuangbanna 666303, China

**Keywords:** heavy metal pollution, *Brassica*, post-translational modification, detoxification, molecular breeding

## Abstract

Protein post-translational modification (PTM) is an efficient biological mechanism to regulate protein structure and function, but its role in plant responses to heavy metal stress is poorly understood. The present study performed quantitative succinyl-proteome profiling using liquid chromatography–mass spectrometry analysis to explore the potential roles of lysine succinylation modification in turnip seedlings in response to cadmium (Cd) stress (20 μM) under hydroponic conditions over a short time period (0–8 h). A total of 547 succinylated sites on 256 proteins were identified in the shoots of turnip seedlings. These succinylated proteins participated in various biological processes (e.g., photosynthesis, tricarboxylic acid cycle, amino acid metabolism, and response to stimulation) that occurred in diverse cellular compartments according to the functional classification, subcellular localization, and protein interaction network analysis. Quantitative analysis showed that the intensities of nine succinylation sites on eight proteins were significantly altered (*p* < 0.05) in turnip shoots after 8 h of Cd stress. These differentially succinylated sites were highly conserved in Brassicaceae species and mostly located in the conserved domains of the proteins. Among them, a downregulated succinylation site (K150) in the glycolate oxidase protein (Gene0282600.1), an upregulated succinylation site (K396) in the catalase 3 protein (Gene0163880.1), and a downregulated succinylation site (K197) in the glutathione S-transferase protein (Gene0315380.1) may have contributed to the altered activity of the corresponding enzymes, which suggests that lysine succinylation affects the Cd detoxification process in turnip by regulating the H_2_O_2_ accumulation and glutathione metabolism. These results provide novel insights into understanding Cd response mechanisms in plants and important protein modification information for the molecular-assisted breeding of *Brassica* varieties with distinct Cd tolerance and accumulation capacities.

## 1. Introduction

Cadmium (Cd), a global pollutant that comes from both anthropogenic (e.g., industrial waste, atmospheric deposition, sewage water, and chemical fertilizers) and natural sources (e.g., parent material, volcanos, fossil fuels) [1,2], is one of the most toxic heavy metals to living organisms [3,4]. In plants, Cd toxicity inhibits carbon fixation, reduces chlorophyll synthesis and photosynthetic activity, causes osmotic stress, and induces overproduction of reactive oxygen species (ROS) to damage the plant membranes and the destruction of the cell organelles [3]. Cd also has harmful effects on the uptake and transportation of mineral elements, resulting in growth inhibition [5]. In agriculture, soil Cd pollution will not only reduce the yield of crops and vegetables, but also pollute these foods [2]. The statistics show that vegetables contribute more than 80% of the total Cd uptake in human bodies [6]. Therefore, crops and vegetables with high capacities for Cd absorption and accumulation are generally objectionable. A variety of agricultural control measures (e.g., agricultural management, application of soil passivators, and foliar spraying of control agents), as well as the breeding of low-Cd-accumulating varieties, have been used to minimize Cd accumulation in crops and vegetables [2]. However, an increasing number of plants with high Cd tolerance and accumulation capabilities have shown potential for remediating Cd-polluted soils [7]. Thus, the breeding (especially molecular-assisted breeding) of low-Cd-accumulating crops and vegetables for food or high-Cd-accumulating plants for phytoremediation is both important and urgent.

The genus *Brassica*, which includes over 150 species of annual, biennial, and perennial herbs, is an important source of vegetables and vegetable oil worldwide. However, a number of *Brassica* species (e.g., *B. juncea*, *B. carinata*, *B. napus*, *B. nigra*, *B. rapa*, and *B. oleracea*) show a certain potential for heavy metal accumulation, which are able to extract a considerable quantity of metals to be successfully used in the phytoremediation process, owing to their large biomass and rapid growth rate [8]. With the development of genome sequencing of the *Brassica* species [9,10], breeding *Brassica* cultivars with different accumulation capacities for toxic metals (especially Cd) for food or phytoremediation by genetic engineering (e.g., genome editing) [11] will be an important future technology to address soil heavy-metal pollution. This requires a thorough understanding of the molecular mechanisms of plant tolerance, absorption, and transport of heavy metals. Among the *Brassica* species, turnip (*Brassica rapa* var. *rapa*) is a widely cultivated vegetable or fodder crop in Asia, Europe, and America. Previous studies found that the turnips cultivated in China had a generally high ability to tolerate and accumulate Cd [12]. The biochemical and molecular mechanisms of Cd tolerance and accumulation in turnips are of research interest and were recently explored using combined biochemical, transcriptomic, and proteomic approaches [13,14,15]. In addition to the conventional Cd transport and detoxification processes, these studies recently reported that the genes and/or proteins involved in the protein acetylation/deacetylation process were obviously induced, and that the overall acetylation modification level increased in turnip shoots under Cd stress [15], which suggests that protein post-translational modifications (PTMs), represented by acetylation in plants, likely play an important role in their response to Cd stress.

The PTMs are efficient biological mechanisms to finely regulate the protein structure, function, localization, activity and protein interaction by covalently binding some small chemical molecular groups at the amino acid side chains and the C- or N-termini of proteins [16,17]. More than 200 PTM types have been identified in organisms [18], and many of the PTM types have been identified during various biological processes, which may constitute a mutual control network [16]. However, a few of the PTMs have been found to be involved in plant responses to heavy metal stress. The lysine acylation modifications (e.g., acetylation, propionylation, butyrylation, succinylation, crotonylation, malonylation, and glutarylation) are common PTMs that often overlap with each other [19,20]. Our latest study indicated that the protein acetylation/deacetylation processes likely participated in the Cd response in the turnip leaves [15], and a previous study reported that oxidative stress triggered interactions between lysine acetylation and succinylation in rice leaves [19]. Oxidative stress is a major symptom of heavy metal toxicity in plants; this study thereby also indicated that acetylation and succinylation may be involved in the plant response to heavy metal stress.

To clarify whether these lysine acylation modifications are involved in the plant’s response to Cd stress and explore the potential mechanisms, the global modification levels of lysine acetylation, succinylation, and crotonylation in the shoots of turnip seedlings exposed to Cd were determined using Western blotting (WB) analysis in this study. The results showed that the changes in succinylation modification were more notable than the changes in acetylation and crotonylation modifications. Therefore, the quantitative succinyl-proteome profiling was further performed to examine the roles of succinylation modification in the response of turnip to Cd stress. The present study provides original information on global protein succinylation modification in Brassicaceae species and elucidates the potential functions of succinylation modification in plant responses to Cd stress.

## 2. Materials and Methods

### 2.1. Plant Material and Cd Treatment

The potential Cd-hyperaccumulator, turnip landrace KTRG-B14 [12], was used in this study, in which the protein acetylation/deacetylation processes were found to participate in Cd response [15]. The seeds were surface sterilized and germinated on petri dishes with double-layer filter papers, as previously described [14]. Five days later, germinated seedlings of uniform size were transplanted into hydroponic disks (l: 31.5 cm, w: 22.0 cm, h: 6.0 cm) filled with 1.5 L 1/2 modified Hoagland nutrient solutions (Figure 1A). Each disk was transplanted with 150 seedlings (Figure 1A), and three hydroponic disks were prepared. The nutrient solutions were renewed every 2 days. Six days later, 20 μM Cd^2+^ (CdCl_2_•2.5H_2_O) was added to the solutions, and the roots and shoots (including cotyledons and leaves) of the turnip seedlings treated with Cd for 0, 0.5, 1, 2, 4, and 8 h were separately harvested. The treatment time was chosen according to a previous experiment [14], in which the expression of numerous genes in turnip roots was significantly changed after Cd treatment for 6 h under a similar experimental method. The samples in three disks for each treatment time were harvested separately as three biological replicates.

### 2.2. Cd Concentration Determination

The shoot and root samples were oven-dried at 80 °C for 48 h and used to detect Cd concentrations using the ICP–MS method, as previously reported [21]. In brief, approximately 0.2 samples were digested with 5 mL of HNO_3_ in polytetrafluoroethylene digestion tanks using the following procedure: 100 °C, 3 min; 140 °C, 3 min; 160 °C, 3 min; 180 °C, 3 min; and 190 °C, 15 min. After cooling, the digestion solutions were transferred to 50-mL volumetric flasks and the volumes were fixed to the measurement scale. Sample solutions were detected using an inductively coupled plasma mass spectrometer (Thermo Fisher Scientific, Waltham, MA, USA) and the Cd concentrations were calculated according to the standard curve, which was drawn by 0, 0.1, 0.2, 0.4, and 1 mg L^−1^ Cd standard solutions.

### 2.3. Protein Extraction, Qualification, and Quantification

The protein extraction was conducted according to previous studies with some modifications [22,23]. In brief, the shoot samples of turnip plants collected at different time points were first ground with liquid nitrogen, then the powders were transferred to a 5 mL centrifuge tube and sonicated three times on ice using a high intensity ultrasonic processor (Scientz) in lysis buffer [22]. Equal volumes of Tris-saturated phenol (pH 8.0) were added and then the mixtures were further vortexed for 5 min, after centrifugation (5000× *g*, 4 °C, 10 min), the upper phenol phases were transferred to a new centrifuge tube. The proteins were precipitated by adding at least four volumes of ammonium sulfate-saturated methanol and incubated at −20 °C for at least 6 h. After centrifugation at 4 °C for 10 min, the supernatants were discarded. The remaining precipitates were washed with ice-cold methanol, followed by ice-cold acetone for three times [23]. The proteins were redissolved in 8 M urea. The quality of protein extraction was detected using sodium dodecyl sulfate-polyacrylamide electrophoresis (SDS–PAGE), and the protein concentrations were determined using a bicinchoninic acid (BCA) kit (Sigma-Aldrich, St. Louis, MI, USA), according to the manufacturer’s instructions.

### 2.4. WB Analysis

Fifteen micrograms of total protein from each sample were used for WB analysis, which was performed as previously described [15]. The polyvinylidene difluoride (PVDF) membranes were incubated with primary antibodies (anti-acetyllysine antibody, anti-succinyllysine antibody, and anti-crotonyllysine antibody: 1:1000 dilution) at 4 °C with gentle shaking overnight. After washing for 30 min with a 1× Tris-buffered saline with Tween-20 (TBST) solution, the membrane was incubated with a horseradish peroxidase (HRP)-conjugated secondary antibody (1:1000 dilution) for 1 h at room temperature. After washing for 30 min with a 1× TBST solution, the chemiluminescence signals of the membranes were detected using an enhanced chemiluminescent (ECL) kit (GE, Evansville, IN, USA). SDS–PAGE gels stained with Coomassie brilliant blue were used to determine the sample loading control. The relative fluorescence intensity of the main bands on the WB image was quantified using ImageJ software [24].

### 2.5. Succinyl-Proteome Profiling

Based on the WB results, quantitative succinyl-proteome profiling was performed to understand the potential mechanisms of the turnip response to Cd. The main experimental and analytical processes (Appendix A), including trypsin digestion, succinylation modification enrichment, liquid chromatography–mass spectrometry (LC–MS) analysis, database searching, motif analysis, and protein function annotation, were performed using previously described methods [22,23,25,26].

#### 2.5.1. Trypsin Digestion

The same amount of protein for each sample was adjusted to the same volume with lysate for trypsin digestion. The samples were slowly added to the final concentration of 20% *v*/*v* TCA to precipitate protein, then vortexed to mix and incubated for 2 h at 4 °C. The precipitates were collected by centrifugation at 4500× *g* for 5 min at 4 °C. The precipitated proteins were washed with pre-cooled acetone for three times and dried for 2 h. The protein samples were then redissolved in 100 mM TEAB and ultrasonically dispersed. The trypsin was added at 1:50 trypsin-to-protein mass ratio for the first digestion overnight. The samples were reduced with 5 mM dithiothreitol for 60 min at 37 °C and alkylated with 11 mM iodoacetamide for 45 min at room temperature in darkness. Finally, the peptides were desalted by C18 SPE column.

#### 2.5.2. Succinylation Modification Enrichment

To enrich succinyl-modified peptides, tryptic peptides dissolved in NETN buffer (100 mM NaCl, 1 mM EDTA, 50 mM Tris-HCl, 0.5% NP-40, pH 8.0) were incubated with pre-washed antibody beads (Lot number 105032317G009, PTM Bio, Chicago, IL, USA) at 4 °C overnight with gentle shaking. Then, the beads were washed for four times with NETN buffer and twice with H_2_O. The bound peptides were eluted from the beads with 0.1% trifluoroacetic acid. Finally, the eluted fractions were combined and vacuum-dried. For liquid chromatography–mass spectrometry analysis (LC–MS/MS) analysis, the resulting peptides were desalted with C18 ZipTips (Z720046, Millipore, Burlington, MA, USA), according to the manufacturer’s instructions.

#### 2.5.3. LC–MS/MS Analysis

The tryptic peptides were dissolved in solvent A (0.1% formic acid and 2% acetonitrile in water) and then separated using NanoElute ultra high performance liquid system. The peptides were separated with a gradient from 5% to 22% solvent B (0.1% formic acid in 90% acetonitrile) over 42 min, 22% to 30% solvent B in 12 min and climbing to 80% in 3 min then holding at 80% for the last 3 min, all at a constant flowrate of 300 nL/min on an EASY-nLC 1200 UPLC system (Thermo Fisher Scientific, Waltham, MA, USA).

The separated peptides were implanted into capillary ion source for ionization then analyzed in timsTOF Pro MS (Bruker Daltonics, Billerica, MA, USA). The ion source voltage applied was 1.6 kV. The peptide parent ions and their secondary fragments were detected and analyzed by high-resolution TOF. The scanning range of the secondary mass spectrometry was set to 400–1500 m/z. The data acquisition mode uses parallel cumulative serial fragmentation (PASEF) mode. After a primary MS is collected, the PSEF mode is used for 10 times to collect the secondary spectrum with the charge number of parent ions in the range of 0–5. The dynamic exclusion time of tandem MS scanning was set to 30 s to avoid repeated scanning of parent ions.

All of the MS data were deposited in the ProteomeXchange Consortium via the PRIDE [27] partner repository with the dataset identifier PXD028098.

#### 2.5.4. Database Searching

The resulting MS/MS data were processed using the MaxQuant search engine (v.1.6.15.0). The data were searched against the turnip (the data are not public yet) and the *B. campestris* genomic database (Blast_Brassica_campestris_3711_TX_H700_genomic_20201221.fasta; 55950 entries) concatenated with a reverse decoy database to calculate the false positive rate (FDR) caused by random matching, and a common contaminated library was added to the database to eliminate the influence of the contaminated proteins in the identification results. The trypsin digestion mode was set to trypsin/P; the number of missing bits was set to four; the minimum length of the peptide was set to seven amino acid residues; the maximum number of peptide modifications was set to five; the mass error tolerance of the primary parent ions in the first search and main search were set to 20 ppm and 20 ppm, respectively; and the mass error tolerance of the secondary fragment ions was set to 20 ppm. The cysteine carbamidomethyl (C) was set as a fixed modification and the variable modifications were set as oxidation of methionine, acetylation of protein N-terminal, and succinylation of lysine.

In order to obtain high-quality analysis results, the database search results need to be further filtered. The FDR for the identification accuracy at the three levels of spectrum, peptide, and protein was set as 1% and the identified protein needs to contain at least one unique peptide.

#### 2.5.5. Motif Analysis

The motif characteristics of the modification sites were analyzed by MoMo analysis tool based on the motif-x algorithm. The peptide sequence, composed of 10 amino acids upstream and downstream of all of the potential and identified modification sites, was used as the analysis background and object, respectively. Those characteristic sequence forms with the number of peptide segments >20 and the statistical test *p* value < 0.000001 were designed as the motifs of the modified peptide segments. Based on the results of the MoMo analysis, the frequency and degree of amino acids appearing near the modification sites were displayed in the form of a heatmap.

#### 2.5.6. Protein Function Annotation

In order to deeply understand the functional characteristics of different proteins, we annotated the identified proteins in an all-round way, including gene ontology (GO) terms, subcellular localization, the Kyoto Encyclopedia of Genes and Genomes (KEGG) pathway, and the protein domain. We analyzed the enrichment of all of the identified proteins at the two levels of KEGG pathway and protein domain by Fisher’s exact test, in order to find out whether the succinyl-modified proteins have a significant enrichment trend in some of the functional types and protein categories.

#### 2.5.7. Protein Interaction Network Analysis

The identified protein database number or protein sequence was compared with the STRING (v.11.0) protein network interaction database [28], and the protein interaction relationship was extracted according to the confidence score >0.7 (high confidence). Then, the protein–protein interaction (PPI) network was visualized by Cytoscape 3.7.2 software [29] and classified according to the KEGG annotation results.

### 2.6. Identification of Differentially Succinylated Sites (DSSs)

The results of the database search and analysis of the MS data gave the signal intensity of each peptide in different samples. According to this information, the relative quantification of the protein was calculated through the following steps: Firstly, after the signal intensity value (*I*) of the peptide in different samples was changed by centralization, the relative quantification value (*R*) of the peptide in different samples was obtained. The calculation formula was as follows: where *i* and *j* represent the sample and the peptide, respectively:
*R_ij_* = *I_ij_*/*Mean* (*Ij*)In order to eliminate the systematic error of sample loading in MS determination, the relative quantitative value of the peptide needed to be corrected by median normalization (*NR*). The calculation formula was as follows:*NR_ij_* = *R_ij_*/*Median* (*R_i_*)The relative quantitative value of the protein was expressed by the median of the relative quantitative value of the corresponding specific peptide to protein. The calculation formula was as follows: where *k* and *j* represent the protein and the corresponding specific peptide of the protein:*R_ik_* = *Median* (*NR_ij_*, *j* ∈ *k*)

The difference analysis of the succinylated sites first calculated the quantitative average values of the modification sites of repeated samples for the samples to be compared, and finally calculated the fold change (*FC*) of the comparison group (8 h) compared with the control group (0 h). The calculation formula was as follows: where *R* represented the relative quantification of the modification site, *i* represented the sample, and *k* represented the modification site:*FC*8 h/0 *h,k* = *Mean* (*R_ik_*, *i* ∈ 8 h)/*Mean* (*R_ik_*, *i* ∈ 0 h)

The significant difference was analyzed by *t*-test at *p* < 0.05 level. The differential succinylated sites were identified on the basic of *FC* > 1.5 or <0.67 and *p* < 0.05.

### 2.7. Analysis of Protein Sequence Characteristics

The homologous sequences of the differential succinylated proteins (DSPs) in several of the cruciferous species, including *Brassica rapa* (genome edition number: CAAS_Brap_v3.01), *B. napus* (genome edition number: Bra_napus_v2.0), *B. oleracea* var. *oleracea* (genome edition number: BOL; cultivar: TO1000), *Raphanus sativus* (genome edition number: Rs1.0; cultivar: WK10039), and *Arabidopsis thaliana* (genome edition number: TAIR10.1), were downloaded from the NCBI database. The homologous protein sequences were aligned by the Clustal W software [30] and visualized by the GeneDoc software to compare the conservativeness of the succinylated sites. The conserved domains of DSPs were searched by Pfam tool (http://pfam.xfam.org/search#tabview=tab1, accessed on 27 July 2021), and then the domain diagrams were drawn using the TBtools software [31].

### 2.8. Enzyme Activity Assay

The activities of glycolate oxidase (GOX), catalase (CAT), and glutathione S-transferase (GST) were measured using their corresponding assay kits (Solarbio, Beijing, China), in accordance with the manufacturers’ instructions [14,15]. To detect the GOX activity, approximately 0.1 g of sample was homogenized in 1 mL of extracting solution on ice and then centrifuged (10,000× *g*, 4 °C, 10 min). The sample supernatant was mixed with GOX working solution at 25 °C and the absorbance of the mixture was determined under 324 nm wavelength. To detect the CAT activity, approximately 0.1 g of sample was homogenized in 1 mL of acetone on ice and then centrifuged (8000× *g*, 4 °C, 10 min). The sample supernatant was mixed with the CAT working solution at 25 °C and the absorbance of the mixture was determined under 240 nm wavelength. To detect the GST activity, approximately 0.1 g of sample was homogenized in 1 mL of extracting solution on ice and then centrifuged (8000× *g*, 4 °C, 10 min). The sample supernatant was mixed with the GST working solution at 25 °C and the absorbance of the mixture was determined under 340 nm wavelength. The active units were defined as follows: GOX—1 nmol glyoxylate phenylhydrazone; CAT—1 nmol H_2_O_2_; and GST—1 nmol 1-chloro-2.4-dinitrobenzene. All of the units were determined by the amount produced, degraded, or catalyzed from 1 g of sample in one min at 25 °C.

### 2.9. Hydrogen Peroxide (H_2_O_2_) Concentration Detection

The H_2_O_2_ concentration in the shoots of turnip seedlings was measured using an H_2_O_2_ assay kit (Solarbio, Beijing, China), in accordance with the manufacturer’s instructions [15]. In brief, approximately 0.1 g of sample was homogenized in 1 mL of acetone on ice and then centrifuged (8000× *g*, 4 °C, 10 min). The sample supernatant and H_2_O_2_ standard solution (1 mmol mL^−1^) were mixed with reaction solutions at 25 °C, and the absorbance of the mixture was measured under 415 nm wavelength.

### 2.10. Statistical Analysis

The bar charts were generated using SigmaPlot 10.0 (SSI, San Jose, CA, USA) and statistical analyses were performed using SPSS version 18.0 (IBM, Chicago, IL, USA). An independent-sample *t* test and one-way ANOVA were used to analyze significant differences between pairs of samples and different samples (≥3), respectively.

## 3. Results

### 3.1. Protein Acylation Levels in Turnip Shoots under Cd Stress

This study analyzed the early (0–8 h) Cd response characteristics of turnip seedlings (Figure 1A). The Cd concentrations in the shoots or roots of turnip seedlings significantly increased from 0.5 h (0.40 ± 0.03 mg kg^−1^ in the shoots and 2.18 ± 0.40 mg kg^−1^ in the roots) to 8 h (10.95 ± 1.11 mg kg^−1^ in the shoots and 26.91 ± 1.04 mg kg^−1^ in the roots) under 20 µM Cd treatment (Figure 1B).

An immunoblot analysis was performed using lysine acetylation-specific, succinylation-specific, and crotonylation-specific pan antibodies to understand whether different acylation modifications were involved in the response of turnip to Cd stress. As shown in Figure 1D–F, lysine acetylation, succinylation, and crotonylation modifications in the proteins were detected in turnip shoots with different molecular weights under different Cd treatment times. The quantitative analysis results showed differential changes in the modification intensities of acetylation, succinylation, and crotonylation in the shoots of turnip seedlings with increasing Cd treatment time (Appendix A). The succinylation modification showed more notable changes in the main bands than those in the acetylation and crotonylation modifications, especially between 0 h- and 8 h-treated samples (Appendix A). Thus, the global lysine succinylation sites at the proteomic level (namely, the succinyl-proteome) were further detected to explore the roles of succinylation modification in the response of turnip to Cd stress in this study.

### 3.2. Global Lysine Succinylation Modification in Turnip Shoots

#### 3.2.1. Identification of Lysine Succinylation in Turnip Shoots

A total of six samples with 0-h and 8-h Cd treatments were used for these experiments. The succinylated peptides and proteins were identified using LC–MS/MS. Only the succinylated sites that were repeatedly identified in at least two independent samples were selected for the bioinformatics analysis to characterize the succinylome in turnip shoots. The detailed identification results are provided in Figure 2A. A total of 547 succinylation sites corresponding to 256 proteins were identified (Figure 2A; Appendix A). The protein sequences of the 256 succinylated proteins are provided in the Supplementary Protein Sequences. More than half of the succinylated proteins contained only one succinylated site, and the number of proteins containing two, three, four, and five succinylated sites accounted for 17.2%, 9.4%, 6.6%, and 3.9% of all of the succinylated proteins, respectively (Figure 2B). A total of 15 proteins contained more than five succinylated sites (Figure 2B). Two proteins in particular, the large chain ribulose bisphosphate carboxylase (RubBisCO; Gene0333600.1) and the dihydrolipoyl dehydrogenase (Gene0496440.1), had the most succinylated sites (13) (Appendix A).

#### 3.2.2. Site Properties of the Succinylated Peptides

To exhibit the features of the identified succinylated peptides, motif analysis was performed using the Motif-X program, which computes the probability analysis of the over-represented or under-represented amino acids adjacent to the succinylation sites. No consensus sequence motif was extracted in this study, which may be attributed to multiple reasons. However, the heatmap analysis of the amino acid sequences around the succinylated lysine sites still showed the presence frequency of the specific amino acids adjacent to the lysine succinylation sites. As shown in Appendix A, the presence frequency of glycine (G) at the +4, +2, +1, −3, and −6 positions was relatively higher, and alanine (A) was relatively over-represented at the +9, +5, and −10 positions. The lysine (K) and valine (V) were over-represented at the +9 and −1 positions, respectively, and isoleucine (I), lysine (K), and arginine (R) were under-represented at the +9, +1, and +1 positions, respectively (Appendix A).

#### 3.2.3. Functional Classification and Subcellular Location Analysis

The succinylated proteins were annotated in multiple ways (Appendix A), including the GO classification, subcellular localization, KEGG pathway, and protein domain, to understand their functional characteristics. The GO classification analysis in the category biological process showed that most of the succinylated proteins were involved in the cellular processes (25.1%), metabolic processes (23.0%), and response to stimulus (21.6%), and the remaining major biological processes were related to biological regulation (7.8%), multi-organism processes (4.7%), developmental processes (4.5%), and multicellular organismal processes (4.3%) in turnip shoots (Figure 3A; Appendix A). The molecular function classification results showed that the major succinylated proteins were related to binding (45.0%), catalytic activity (33.7%), and structural molecule activity (11.3%) in turnip shoots (Figure 3B; Appendix A). For the cellular component analysis, succinylated proteins were distributed in the cell (43.3%), intracellular (43.1%), and protein-containing complex (13.6%) (Appendix A).

The subcellular location analysis showed that the dominant succinylated proteins were located in the chloroplasts (39.1%), cytoplasm (28.1%), and mitochondria (18.0%) in turnip shoots (Figure 3C; Appendix A). A small proportion of the succinylated proteins were predicted to be localized in the nucleus (6.6%), plasma membrane (3.5%), and cytoskeleton (2.0%) (Figure 3C; Appendix A).

The functional enrichment analysis, based on the KEGG pathways and protein domains, was further performed to reveal the characteristics of the succinylated proteins in turnip plants in greater detail. A total of 21 significantly enriched pathways were identified using KEGG pathway enrichment analysis (Figure 3D; Appendix A). The glyoxylate and dicarboxylate metabolism was the most significantly enriched pathway, followed by the tricarboxylic acid (TCA) cycle and carbon fixation in photosynthesis (Figure 3D; Appendix A). Several energy production- and glucometabolic-related pathways, including glycolysis/gluconeogenesis, pyruvate metabolism, and oxidative phosphorylation, and two photosynthesis-related pathways, photosynthesis and photosynthesis-antenna proteins, were also significantly enriched (Figure 3D; Appendix A). The proteins belonging to the ribosome and peroxisome were also frequently modified by succinylation in turnip shoots (Figure 3D; Appendix A). Most of the remaining highly-enriched pathways were related to the metabolism or biosynthesis of diverse metabolites (Figure 3D; Appendix A), such as nitrogen, propanoate, selenocompound, folate, ascorbate, aldarate, and various amino acids (e.g., glycine, serine, threonine, alanine, aspartate, glutamate, valine, leucine and isoleucine, cysteine, methionine, arginine, and lysine).

A total of 39 significantly enriched protein domains were identified (Appendix A), and the top 20 enriched domains are represented with a bar chart (Figure 3E). The results indicated that the top two significantly enriched domains were related to the glycine cleavage H-protein and biotin-requiring enzyme (Figure 3E; Appendix A). The domains related to transketolase, ATP synthase, elongation factor Tu, and cobalamin-independent synthase were also highly enriched (Figure 3E; Appendix A). Most of the remaining highly enriched domains were related to proteins involved in photosynthesis (photosystem I reaction center subunit IV, fructose-bisphosphate aldolase, carbonic anhydrase, and ribulose bisphosphate carboxylase small chain), the TCA cycle (aconitate hydratase), glycolysis (triosephosphate isomerase), redox (peroxiredoxin), and ribosome (ribosomal protein L6) proteins (Figure 3E; Appendix A). These enriched protein categories were consistent with the significantly enriched KEGG pathways (Figure 3D).

#### 3.2.4. PPI Network Construction

The PPI network analysis was performed to investigate the interactions between different succinylated proteins. A total of 167 proteins carrying various succinylation sites were matched to the PPI network (Figure 4; Appendix A). Most of the succinylated proteins were clustered into two highly interconnected networks, ribosome and photosynthesis (Figure 4). The two inter-networks were linked to each other via some of the succinylated proteins (Figure 4), which suggests that succinylation regulates the crosslinks between these biological processes. The computed node degrees for all of the succinylated proteins are provided in Appendix A. The results showed that there were one, five, and forty-one succinylated proteins with degrees over 50, 40, and 30 in the network, respectively. Some of the important nodes with relatively higher degrees contained multiple succinylation sites in the network, such as phosphoglycerate kinase (eight sites), glyceraldehyde-3-phosphate dehydrogenase (seven sites), glyceraldehyde-3-phosphate dehydrogenase (six sites), epimerase domain-containing protein (5 sites), and amino methyltransferase (five sites) (Appendix A). The constructed PPI network information indicated that the lysine succinylation was relatively active in the ribosome and photosynthesis processes in the shoots of turnip seedlings, which conformed to the KEGG enrichment analysis results (Figure 3D).

### 3.3. Differentially Succinylated Sites (DSSs) in Turnip Shoots in Response to Cd Stress

#### 3.3.1. Identification of DSSs

To reveal the potential role of succinylation modification in the turnip response to Cd stress, a quantitative analysis of the succinylated sites between Cd-treated (8 h) and control (0 h) samples was performed in this study.

A total of 263 succinylated sites were successfully quantified in different samples (Figure 2A; Appendix A), and nine DSSs (five upregulated and four downregulated) on eight proteins were identified (Table 1; Appendix A), which represented a 1.5-fold change in the modification intensity with *p* < 0.05. Except for the 60S ribosomal protein L9, each of the other seven differentially succinylated proteins (DSPs) contained more succinylated sites than the DSSs (Table 1), which indicated that the succinylation modification of the different sites in the same protein was not regulated synchronously under Cd stress. Notably, each of the DSSs was highly conserved among the homologous protein sequences of the representative Brassicaceae species (Figure 5A). Furthermore, seven of the nine DSSs were located in the core domains of the proteins (Figure 5B).

These DSPs included five enzymes that were involved in glutamate metabolism, glyoxylate metabolism, peroxisome, glutathione (GSH) metabolism, and the TCA cycle (Figure 6A–D). Three DSPs (i.e., glycolate oxidase, catalase 3, and glutathione S-transferase) are involved in the production and decomposition of H_2_O_2_, and GSH metabolism (Figure 6B,C), which are highly associated with Cd detoxification in plants [15]. Therefore, we further analyzed the potential roles of these DSSs in Cd detoxification in turnip shoots.

#### 3.3.2. Variations in H_2_O_2_ Metabolism under Cd Stress

The activities of GOX and CAT were detected to evaluate the functions of the DSSs identified in the glycolate oxidase protein (Gene0282600.1) and the catalase 3 protein (Gene0163880.1) (Table 1; Figure 6B). As shown in Figure 6E,F, the total activities of the GOX and CAT in turnip shoots were significantly decreased and increased, respectively. Notably, the final H_2_O_2_ accumulation was similar between the 0-h and 8-h Cd-treated samples (Figure 6G). The results suggest that the GOX–CAT cycle process in turnip shoots was important in regulating H_2_O_2_ accumulation under Cd stress.

#### 3.3.3. Variations in GSH Metabolism under Cd Stress

The GST activity was detected to evaluate the function of the downregulated succinylated site located on the glutathione S-transferase protein (Gene0315380.1) (Table 1). As shown in Figure 6H, the GST activity in the Cd-treated (8 h) sample was significantly (*p* < 0.01) increased compared to that in the control (0 h) sample.

## 4. Discussion

### 4.1. Changes in Protein Acylation Levels in Turnip Shoots under Cd Stress

The changes in the Cd concentrations in turnip seedlings in this study (Figure 1B) confirmed that turnip accumulated high amounts of Cd under Cd treatment for several hours [14]. Recent omics techniques (e.g., transcriptomics, proteomics, and metabolomics) have greatly helped elucidate the molecular mechanisms of plant responses to Cd stress [15,32]. In addition to the changes in the abundances of genes, proteins, or metabolites, new insights (e.g., DNA methylation and protein PTMs) will improve our understanding of the regulatory mechanisms of gene or protein functions. Based on two recent studies that indicated the potential roles of lysine acylation modifications (e.g., acetylation and succinylation) in the plant response to Cd stress [15,19], the present study performed an immunoblot analysis to further understand whether different acylation modifications were involved in the response of turnip to Cd stress. Almost all of the detected protein bands modified by acetylation, succinylation, and crotonylation (Figure 1D–F) indicate that lysine acylation modifications are common PTMs in plants. Consistent with several previous studies [20,26,33], the WB results (Figure 1D–F) also suggest that these three acylation modifications likely overlap with each other in turnip. As a result of the more notable change in the succinylation modification intensity with increasing Cd treatment time (Appendix A), we performed quantitative succinyl-proteome profiling to explore the roles of succinylation modification in the response of turnip to Cd stress.

### 4.2. Global Analysis of Lysine Succinylation in Turnip Shoots

Since the first identification of succinylation modification in *Escherichia coli* [34], succinyl-proteome analysis has been performed in diverse organisms, including microorganisms, animals, humans, and plants [35,36,37], which indicates that lysine succinylation regulates various important biological processes in organisms. At least 18 studies examined the succinyl-proteome in approximately 14 seed plant species, which identified 142–5502 succinylated sites on 86–2593 proteins (Appendix A). The number of succinylated sites (547 succinylated sites on 256 proteins) identified in this study was within the median range of the numbers found in previous studies (Appendix A), suggesting that our results are reasonable and reliable. The Gramineae plants including, rice (*Oryza sativa* spp. *japonica* cv. Nipponbare and *O. sativa* cv. ‘Wuyunjing 7’), wheat (*Triticum aestivum* cv. ‘Qing Mai 6’), barley (*Hordeum vulgare*), and *Brachypodium distachyon* were the most commonly studied (in seven studies) using succinyl-proteome profiling (Appendix A). These studies provide important information for exploring the functions of succinylation modification in various biological processes in Gramineae plants. However, succinyl-proteome profiling has not been performed in Brassicaceae plants, even in the model plant *Arabidopsis thaliana*, according to a recent database (http://www.peptideatlas.org/builds/arabidopsis/, accessed on 13 April 2022). The present study provides novel PTM information for understanding the molecular mechanisms of the biological processes and activity in Brassicaceae plants, especially in the *Brassica* species.

The identified succinylation sites and proteins vary greatly between different species, which may be attributed to multiple reasons [38]. However, the global succinyl-proteome characteristics showed some similarity between turnip and other species. For example, the number of succinylated sites per protein was negatively correlated with the corresponding protein number in turnip (Figure 2B), which is consistent with the results in *Broussonetia papyrifera* [22]. Among these succinylated proteins, RuBisCO and dihydrolipoyl dehydrogenase contained as many as 13 succinylated sites (Appendix A). RuBisCO is the most abundant protein involved in carbon assimilation in plant leaves, and it also included the most succinylation sites in some other species [19,38]. The high number of succinylation sites in the dihydrolipoyl dehydrogenase (Appendix A), which is located in mitochondria and participates in energy metabolism [39], represented a specific succinylation feature in the turnip shoots. These observations suggest that succinylation is an important PTM for regulating photosynthetic CO_2_ assimilation and energy metabolism in turnip plants.

In contrast to previous studies [19,22,38], no consensus sequence motif was extracted from the succinylated proteins in this study, which may be attributed to multiple reasons. However, the main over-represented amino acids (i.e., glycine and alanine) surrounding the succinylated lysine sites in the turnip shoots (Appendix A) were consistent with other species [22,38], which may be because these amino acids (e.g., glycine and alanine), with relatively weak steric hindrance effects, are more compatible with succinyl groups [38]. The results indicate the specific and common lysine succinylation characteristics in turnip seedlings compared to other species.

Previous studies reported that the categories of succinylated proteins identified in the different organs and tissues of plants had relatively higher variations [38,40]. Therefore, we compared the functional classification of the identified succinylated proteins in turnip shoots with those in the leaves or shoots of other plant species. As shown in Appendix A, most of the species showed a similar pattern in GO classification, but the detailed proportion of each item was different. The top three biological processes in most of the species were metabolic processes, single-organism processes, and cellular processes (Appendix A). Notably, the item response to stimulus in turnip shoots, rather than the single-organism process, peculiarly occupied a significant proportion (Appendix A). This result may be determined by the biological traits and adaptive evolution of turnip. Unlike the biological process category, the catalytic activity and binding activity were the two highest terms in turnip and other species at the molecular function level (Appendix A). The subcellular localization of succinylated proteins was also compared between turnip and other plant species. As shown in Appendix A, most of the succinylated proteins were located in the chloroplast, cytoplasm, and mitochondria in the shoots or leaves of all of the species, which suggests that the major biological functions of succinylated proteins in plants are similar in different species.

The top three KEGG pathways, including glyoxylate and dicarboxylate metabolism, TCA cycle, and carbon fixation (Figure 3D; Appendix A), enriched by succinylated proteins in turnip were also observed in the shoots or leaves of the most reported species [19,22,38]. Several energy production- and metabolism-related pathways (i.e., pyruvate metabolism, oxidative phosphorylation, and photosynthesis) that were significantly enriched in turnip (Figure 3D; Appendix A) were also frequently enriched in other species [22,38]. These results suggest that the main metabolic pathways regulated by succinylation modification are relatively conserved in plants. However, succinylated proteins and sites belonging to the same pathways are often different between diverse species for multiple reasons [35,38,40]. Three enriched KEGG pathways (i.e., ribosome, nitrogen metabolism, and peroxisome) in turnip shoots (Figure 3D; Appendix A) were relatively less enriched in other species [38,41,42], which suggests the special roles of these pathways in the shoots of turnip seedlings. The cellular compartments where the main enriched KEGG pathways occurred, such as photosynthesis in chloroplasts and the TCA cycle in mitochondria, conformed to the subcellular localization results of the succinylated proteins (Figure 3C). The enriched protein domains in this study (Figure 3E), which were mostly different from the shoots or leaves of other species [22,38], were also in accordance with the KEGG pathways enriched by succinylated proteins (Figure 3D). Previous studies found that Cd stress influenced many of these enriched KEGG pathways in turnips [14,15], suggesting that the succinylation modification in these pathways may play an important regulatory role in the response to Cd stress in turnip.

The PPI network analysis showed a complex interaction between different succinylated proteins (Figure 4). The results indicated that lysine succinylation was relatively active in the ribosome and photosynthesis processes in the shoots of turnip seedlings, which conformed to the KEGG enrichment analysis results (Figure 3D). The results were partially consistent with previous succinyl-proteome profiling in other plant species [38,41], which suggests the specific characteristics of protein succinylation modification in the shoots of turnip seedlings.

In summary, these results suggest that the succinylated proteins in turnip shoots highly interact with each other and participate in various biological processes that occur in diverse cellular compartments.

### 4.3. DSSs in Turnip Shoots in Response to Cd Stress

To reveal the potential role of succinylation modification in the response of turnip to Cd stress, succinylated sites were quantified between Cd-treated (8 h) and control (0 h) samples. The small number of DSSs (five upregulated and four downregulated) identified in this study (Table 1) may have been attributed to the relatively short Cd treatment time. This is because a greater number of succinylated sites showed a certain degree of changes in intensities (Appendix A), which likely changed significantly under more severe Cd stress. The highly conserved DSSs on the homologous protein sequences of the representative Brassicaceae species (Figure 5A), most of which were located on the conserved domains of the proteins (Figure 5B), indicate that these succinylation sites may play important roles in the response to Cd stress in the whole Brassicaceae family, and even larger plant taxa. To date, no succinyl-proteome profiling research has examined the interaction between plants and heavy metals. The present study thereby provides first-hand information on protein succinylation in the plant’s response to heavy metals.

Previous studies indicated that lysine succinylation caused a greater change in the charge on the lysine target residue (from +1 to −1) than some other acylation modifications, which led to more substantial changes in the chemical properties of the target proteins [19]. The lysine succinylation modification has been found to positively or negatively influence the activities of proteinaceous enzymes [19,43]. In this study, the DSSs on the five proteinaceous enzymes involved in glutamate metabolism, glyoxylate metabolism, peroxisome, GSH metabolism, and the TCA cycle (Figure 6A–D) likely greatly improved or limited the related biological reaction processes in the turnip shoots under Cd stress. Because H_2_O_2_ homeostasis n and GSH metabolism are highly associated with Cd detoxification in plants [15], we further analyzed the potential roles of the three DSPs (glycolate oxidase, catalase 3, and glutathione S-transferase) that participate in these biological processes (Figure 6B,C) in Cd detoxification in turnip shoots.

H_2_O_2_ is a significant regulatory component that is related to signal transduction in plants [44], but an excessive accumulation of H_2_O_2_ and other ROS is harmful to plants under Cd stress. For Cd-tolerant plants, the scavenging of excess ROS in a timely manner via the antioxidant system is an important Cd detoxification strategy. Peroxisomes are the most important sites of H_2_O_2_ production in plants, especially in C_3_ plants [45]. The glycolate produced during photorespiration may be transferred from chloroplasts into peroxisomes to produce glyoxylate and H_2_O_2_ under the catalysis of GOXs [45]. In contrast, the CATs catalyze the decomposition of H_2_O_2_ into H_2_O and O_2_ in peroxisomes [46]. The induction of the CAT activity to control H_2_O_2_ accumulation is an essential mechanism for Cd detoxification in turnip [15], which suggests a confrontation between the regulation GOX and CAT activities during Cd stress. Notably, a downregulated (K150) and an upregulated succinylated site (K396) were identified in a GOX protein (glycolate oxidase) and a CAT protein (catalase 3) in the Cd-treated (8 h) sample compared to the control (0 h) sample in this study, respectively (Table 1; Figure 6B). Because the DSSs were located in the conserved domains of both of the enzymes (Figure 5B), the activity of these two enzymes was likely to be regulated by the changes in the intensities of the succinylation sites, according to a previous study, in which the succinylated K150 site negatively regulated OsCATA activity in rice [19]. The total activities of the GOX and CAT in turnip shoots were significantly altered in this study (Figure 6E,F), and the increased CAT activity was consistent with previous studies [15]. The results suggest that the GOX–CAT cycle process in turnip shoots is helpful in regulating H_2_O_2_ accumulation under Cd stress. As a result, the final H_2_O_2_ accumulation was effectively controlled (Figure 6G), to avoid oxidative stress in plant cells. The alterations in the GOX and CAT activities may have been partially attributed to the succinylation modification regulation of the GOX and CAT members identified in this study. However, the existing experimental system for turnip hindered direct verification of the positive or negative effects of the DSSs on these enzyme activities.

The GSH metabolism is another key pathway that turnips use to detoxify Cd [14,15]. GSH conjugates with Cd to reduce Cd toxicity under the catalysis of GSTs [47]. The increased GST activity in the Cd-treated (8 h) sample in this study (Figure 6H) was consistent with a previous study [14]. Although the upregulated expression of the GST proteins may be an important reason for the improved GST activity [14,15], protein PTMs (e.g., succinylation) also likely affect GST activity. For example, a previous study reported that the relative intensities of several succinylation sites on the OsGSTU6 protein in rice were positively correlated with its activity [19]. Notably, this study identified a downregulated succinylated site (K197) in a GST protein (glutathione S-transferase) (Table 1), which suggests that the decreased intensity of this succinylation site likely influences GST activity, even when the DSS was not located on the conserved domain (Figure 5B). However, whether the intensity of the succinylation site is positively or negatively correlated with the activity of the GST protein requires further validation.

In summary, the quantitative succinyl-proteome analysis identified nine DSSs on eight proteins in 8-h Cd-treated turnip shoots. These DSSs were highly conserved in cruciferous plants, and most of them were located on the core domains of the proteins, which suggests they play important roles in the plant’s response to Cd stress. The changes in the intensities of these succinylated sites on the members of GOXs, CATs, and GSTs likely affected the activities of these proteinaceous enzymes, which contributed to the Cd detoxification in turnip.

## 5. Conclusions

This study explored the potential roles of lysine acylation modification in turnip seedlings exposed to Cd stress. Based on the changes in the levels of acetylation, succinylation and crotonylation modifications detected by WB under 20 μM Cd treatment for a short time (0–8 h), quantitative succinyl-proteome profiling was performed in the shoots of turnip seedlings. A total of 547 succinylation sites on 256 proteins were successfully identified, which provided useful information to study the regulatory mechanism of protein succinylation on the biological processes of Cruciferae plants. Motif analysis, GO classification, subcellular localization, KEGG-based and domain-based enrichment analysis, and protein interaction network analysis indicated that the succinylation sites highly interacted with each other and participated in various biological processes that occurred in diverse cellular compartments. The results showed specific and common succinylation characteristics in turnip shoots, compared to other plant species. The quantitative analysis showed that the intensities of nine conserved succinylation sites on eight proteins in the shoots of turnip seedlings were significantly altered after 8 h of Cd stress. Most of these DSSs were located in the cored domains of the proteins, which suggests that the changes in the modification levels of these succinylation sites greatly affect the protein functions or enzyme activities. Notably, a downregulated succinylation site (K150) in a GOX protein and an upregulated succinylation site (K396) in a CAT protein involved in H_2_O_2_ metabolism, as well as a downregulated succinylation site (K197) in a GST protein, likely affected the activity of the corresponding enzymes and regulated the Cd detoxification process in turnip plants. This study provides novel insights into understanding the Cd response mechanisms in plants and the results are conducive to breed *Brassica* cultivars with a distinct Cd tolerance and accumulation capacities by editing codons of the target lysine succinylation sites. However, several related studies are worth further investigating in the future. For example, the specific effects of identified succinylation sites on protein functions or enzyme activities need further experimental verification. In addition, the other acylation modifications (e.g., acetylation and crotonylation) of plants in response to Cd stress should also be identified, and the overlap of different acylation modifications needs to be resolved.

## Figures and Tables

**Figure 1 cells-11-01947-f001:**
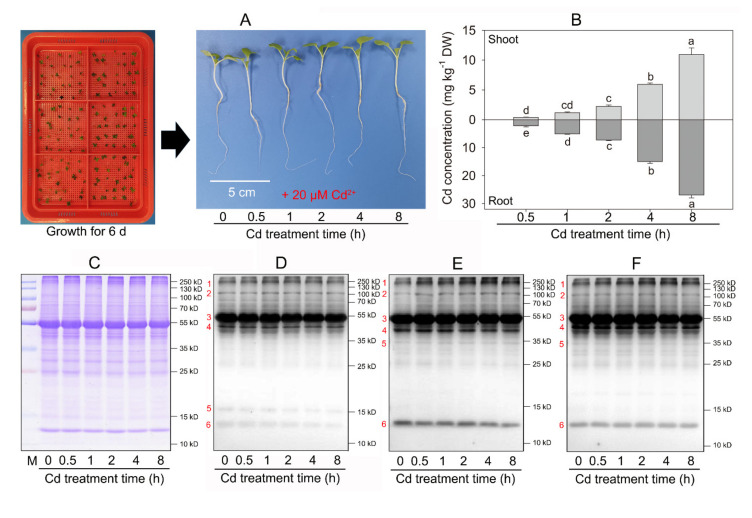
Plant seedling treatment, Cd concentration, and Western blotting (WB) analysis of three acylation modifications of turnip under Cd treatment (20 µM) for different times. (**A**) Treatment mode and morphology of plant seedlings; (**B**) Cd concentrations in turnip seedlings. Data represent means ± standard deviations (*n* = 3); bars labeled with different letters indicate significant differences (*p* < 0.05) between different samples; (**C**) A representative SDS–PAGE gel stained by Coomassie brilliant blue to reflect the protein loading control (15 µg lane^−1^); (**D**) WB results of acetylation modification; (**E**) WB results of succinylation modification; (**F**) WB results of crotonylation modification. The relative fluorescence intensity of the bands labeled with Arabic numbers is shown in the Appendix A, Appendix A.

**Figure 2 cells-11-01947-f002:**
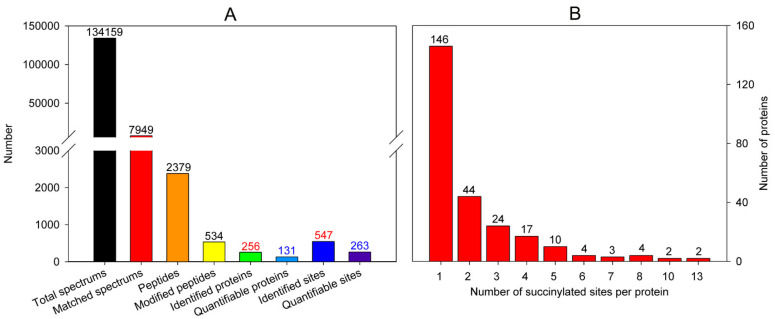
Statistics of basic LC–MS/MS data (**A**) and identified proteins with different numbers of succinylated sites (**B**) in the shoots of turnip seedlings.

**Figure 3 cells-11-01947-f003:**
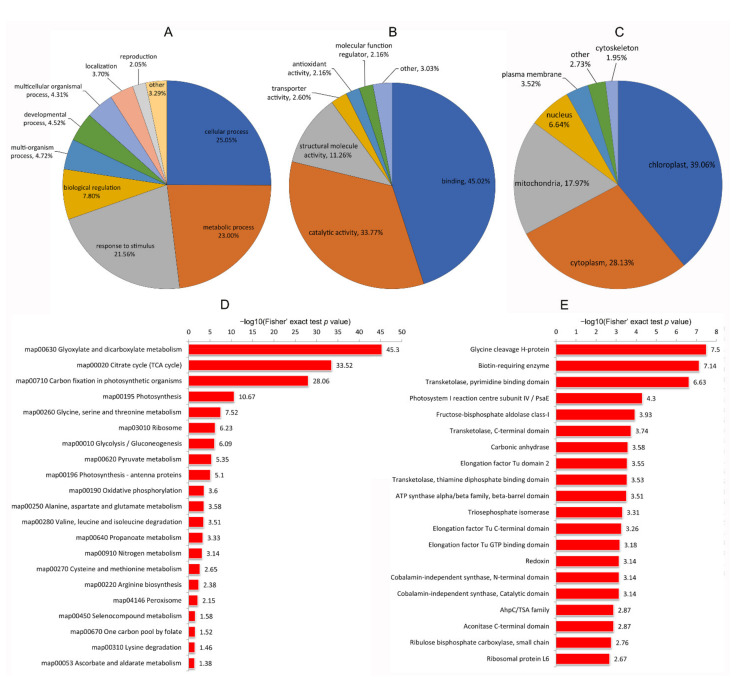
Classification and subcellular location analysis of succinylated proteins identified in the shoots of turnip seedlings. (**A**) GO-based biological process classification of succinylated proteins; (**B**) GO-based molecular function classification of succinylated proteins; (**C**) Subcellular location analysis of succinylated proteins; (**D**) Enriched KEGG pathways of succinylated proteins; (**E**) Top 20 enriched domains of succinylated proteins.

**Figure 4 cells-11-01947-f004:**
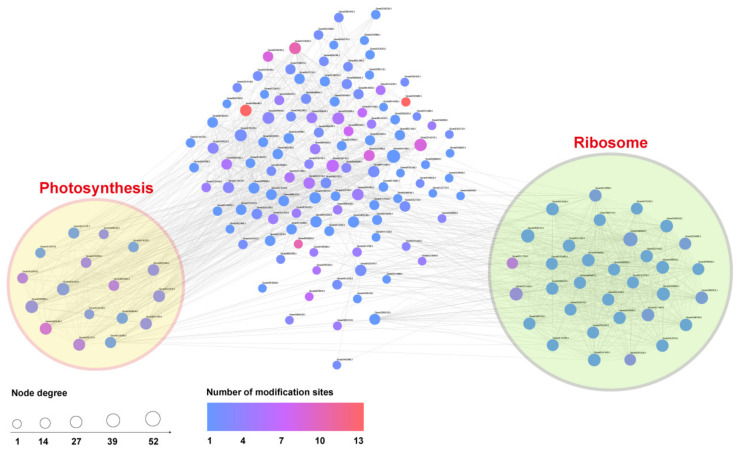
Protein interaction networks of succinylated proteins identified in the shoots of turnip seedlings. The circle size represents the node degree, and the color represents the succinylation site number on the nodes.

**Figure 5 cells-11-01947-f005:**
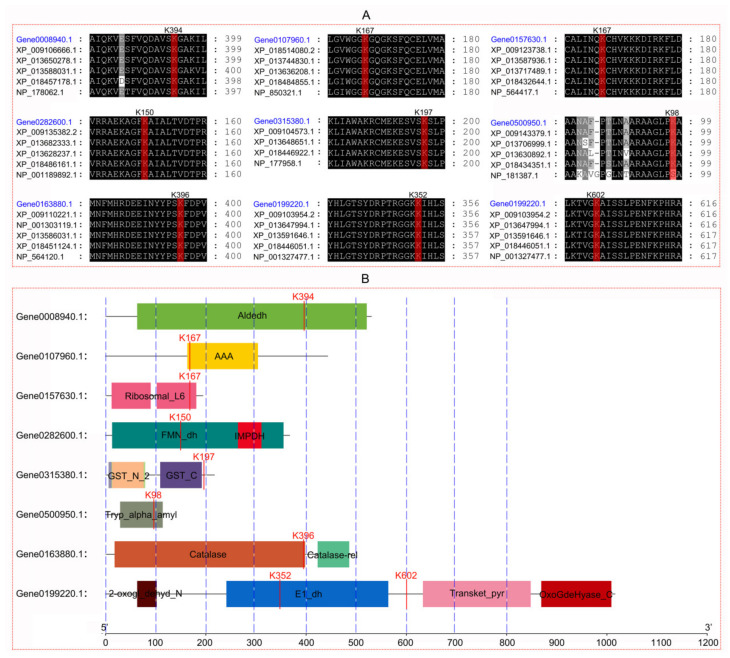
Conservatism and location analysis of the differentially succinylated sites (DSSs) identified in the shoots of turnip seedlings under Cd treatment. (**A**) Alignment of the DSSs with homologous protein sequences in different Brassicaceae plants. The identifiers following the target protein accession (in blue) from top to bottom represent the accession No. in NCBI of the corresponding homologous proteins from *Brassica rapa*, *B. napus*, *B. oleracea* var. *oleracea*, *Raphanus sativus*, and *Arabidopsis thaliana*. The homologous protein from *B. oleracea* var. *oleracea* for Gene0315380.1 is missing in this study; (**B**) The positional relationships of the DSSs with conserved protein domains. The conserved domains were searched using the Pfam tool (http://pfam.xfam.org/search#tabview=tab1, accessed on 27 July 2021).

**Figure 6 cells-11-01947-f006:**
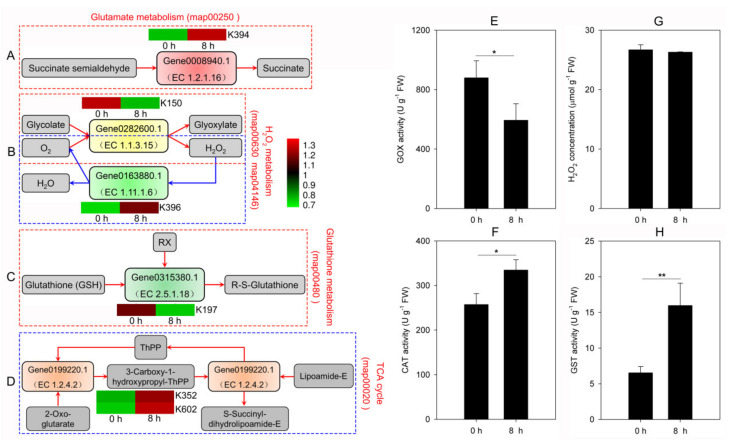
The main reactions catalyzed by the differentially succinylated proteinous enzymes and the corresponding enzyme activity or product concentrations in the shoots of turnip seedlings. (**A**) The reaction catalyzed by succinate-semialdehyde dehydrogenase (Gene0008940.1); (**B**) The reactions catalyzed by glycolate oxidase (GOX; Gene0282600.1) and catalase (CAT; Gene0163880.1); (**C**) The reactions catalyzed by glutathione S-transferase (GST; Gene0315380.1); (**D**) The reactions catalyzed by 2-oxoglutarate dehydrogenase (Gene0199220.1); (**E**) GOX activity in the shoots of turnip seedlings; (**F**) CAT activity in the shoots of turnip seedlings; (**G**) H_2_O_2_ concentration in the shoots of turnip seedlings; (**H**) GST activity in the shoots of turnip seedlings. Data represent means ± standard deviations (*n* = 3); * and ** represent significant differences between pairs of samples at 0.01 < *p* < 0.05 and 0.001 < *p* < 0.01, respectively (**E**–**H**).

**Table 1 cells-11-01947-t001:** Basic information of the differentially succinylated sites identified in the shoots of turnip seedlings in response to Cd stress.

Protein Accession	Protein Description	Subcellular Localization	Protein Size (aa)	Succinylated Site (s)	Differentially Succinylated Site	Fold Change (8 h/0 h)
Gene0008940.1	Succinate-semialdehyde dehydrogenase	mitochondria	530	K394/K442	K394	1.66
Gene0107960.1	ATPase_AAA_core domain-containing protein	chloroplast	443	K147/K167/K 171/K218/K221/K302/K359/K368	K167	0.57
Gene0157630.1	60S ribosomal protein L9	cytoplasm	194	K167	K167	0.66
Gene0282600.1	Glycolate oxidase	cytoplasm	367	K132/K135/K150	K150	0.61
Gene0315380.1	Glutathione S-transferase	chloroplast	217	K52/K197	K197	0.64
Gene0500950.1	Non-specific lipid-transfer protein	extracellular	118	K98/K107/K110	K98	2.01
Gene0163880.1	Catalase 3	peroxisome	492	K396/K481	K396	1.62
Gene0199220.1	2-oxoglutarate dehydrogenase, E1 subunit	mitochondria	1016	K352/K518/K602	K352	1.70
Gene0199220.1	2-oxoglutarate dehydrogenase, E1 subunit	mitochondria	1016	K352/K518/K602	K602	1.55

## Data Availability

The data presented in this study are openly available in the ProteomeXchange Consortium (dataset identifier PXD028098) and the Appendix A.

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
