# Peer review of "Quantitative Succinyl-Proteome Profiling of Turnip (Brassica rapa var. rapa) in Response to Cadmium Stress"

_cells, 2022, doi:10.3390/cells11121947_

Round 1

Reviewer 1 Report

This article presented Quantitative succinyl-proteome profiling of turnip (Brassica rapa var. rapa) in response to cadmium stress. The present study performed quantitative succinyl-proteome profiling to explore the potential roles of lysine succinylation modification in turnip seedlings in response to cadmium (Cd) stress. Before recommending this article for publication, there are some shortcomings for that should be resolve.

General comments

Overall, the study is well designed and presented in a good way, but grammatical and typo mistakes of the manuscript should be revised.

Abstract

The authors elaborated abstract in a good way but the author should mention methods and techniques used in this study that how stress was applied and succinyl was quantified.  

Introduction

The introduction part is well written but needs to add some more information.

Add physical and chemical properties of cadmium by citing the following relevant reference. 10.1002/jemt.23553.

Also add effects of cadmium on plant growth and production.

Provide reasons for cadmium stress.

Discuss the role and potential of Brassica species in heavy metal stress reduction.

How to avoid cadmium contamination?

Extend the literature review on how to study pathways and its associated genes by citing recent literature. 10.3390/ijms22179175, 10.1007/s10725-021-00785-7,

At the end of the introduction aims and objective must be clarifying.

Materials and Methods

As a whole methodology is well written but some points are missing.

Elaborate the methodology of section 2.2.

Section 2.3 references are missing.

Section 2.6 change is to was “protein is calculated’.

As a whole the methodology must be written in past form.

Section 2.9 must be elaborated.

Results and discussion

Results and discussion are well written and presented.

The author must discuss that which molecular mechanisms are necessary to evaluate for further exploration of the combined metals stress on plants. Also discuss role of hormones and associating factors controlling proteins production under cadmium stress conditions.  

Conclusion

The last paragraph must be based on conclusion and future recommendations of the study.

Reviewer 2 Report

The manuscript by Li et al. is well in the scope of the journal and it may contain novel findings that can help to explain the potential roles of lysine acylation modification in turnip seedlings exposed to Cd stress. However, I have two critical questoins:

-Please mention your work aims in the introduction.

-Why did you choose the KTRG-B14 landrace cultivar? What are the characteristics of this cultivar? Post this information in Material and Methods section.

In Figure 2A there is no description of the y axis.

Reviewer 3 Report

Comments and Suggestions for Authors

Line 29: (catalase 3) what does it mean?

Line 117: delete .

Line 118: delete .

Line 147: (Dong et al., 2021; Xu et al., 2017; Zhang et al., 2019; Zhen 147 et al., 2016). Please follow the journal roles (Delete the years and add number instead)

Line 168: Please add the device model and country of manufacture

Line 175: Please add the country of manufacture

Line 264: delete year and add number in the true order

Line 267: please write the method in detail and add the reference

Line 274: provide the method in detail

Line 463: subscript (H2O2)

Line 560: delete year and add number in the true order

Line 574: delete year and add number in the true order

Please follow the Journal roles in the References (see the highlighted comments in the ref.).

Please see the comments in the Pdf version.

Best regards

Round 2

Reviewer 1 Report

good work

Reviewer 3 Report

The paper is accepted in present form